# Role of PI3K-AKT-mTOR Pathway as a Pro-Survival Signaling and Resistance-Mediating Mechanism to Therapy of Prostate Cancer

**DOI:** 10.3390/ijms222011088

**Published:** 2021-10-14

**Authors:** Thanakorn Pungsrinont, Julia Kallenbach, Aria Baniahmad

**Affiliations:** Institute of Human Genetics, Jena University Hospital, 07747 Jena, Germany; thanakorn.pungsrinont@med.uni-jena.de (T.P.); julia.kallenbach@uni-jena.de (J.K.)

**Keywords:** prostate cancer, androgen receptor, castration-resistance, AR antagonist resistance, PI3K, PKB/AKT, mTOR

## Abstract

Androgen deprivation therapy (ADT) and androgen receptor (AR)-targeted therapy are the gold standard options for treating prostate cancer (PCa). These are initially effective, as localized and the early stage of metastatic disease are androgen- and castration-sensitive. The tumor strongly relies on systemic/circulating androgens for activating AR signaling to stimulate growth and progression. However, after a certain point, the tumor will eventually develop a resistant stage, where ADT and AR antagonists are no longer effective. Mechanistically, it seems that the tumor becomes more aggressive through adaptive responses, relies more on alternative activated pathways, and is less dependent on AR signaling. This includes hyperactivation of PI3K-AKT-mTOR pathway, which is a central signal that regulates cell pro-survival/anti-apoptotic pathways, thus, compensating the blockade of AR signaling. The PI3K-AKT-mTOR pathway is well-documented for its crosstalk between genomic and non-genomic AR signaling, as well as other signaling cascades. Such a reciprocal feedback loop makes it more complicated to target individual factor/signaling for treating PCa. Here, we highlight the role of PI3K-AKT-mTOR signaling as a resistance mechanism for PCa therapy and illustrate the transition of prostate tumor from AR signaling-dependent to PI3K-AKT-mTOR pathway-dependent. Moreover, therapeutic strategies with inhibitors targeting the PI3K-AKT-mTOR signal used in clinic and ongoing clinical trials are discussed.

## 1. Introduction

Over decades, prostate cancer (PCa) has been ranked as the most diagnosed malignancy and as the second leading cause of cancer-related mortality of men in many Western countries [1,2,3]. An increased risk of developing PCa is associated with multiple factors including age, heredity, race/ethnicity, and geography. PCa is an endocrine-related disease, in which male hormone androgens play an important role in PCa progression through activation of the androgen receptor (AR) [4,5]. This makes AR a major therapeutic target for PCa therapy. Yet, after a certain period of time with treatment, the tumor will eventually develop a resistance. 

The relapse, or re-progression, of the tumor, usually via adaptive responses, relies on activated alternative pathways besides AR signaling [6,7,8,9,10]. This includes activation of PI3K-AKT-mTOR signaling, which is a well-known pathway that regulates multiple signal transductions and biological processes such as transcription, protein synthesis, metabolism, autophagy, cell proliferation, apoptosis, angiogenesis, migration, etc. [11,12,13,14,15,16]. In this review, we highlight the role of PI3K-AKT-mTOR signaling as a resistance mechanism for PCa therapy in both AR dependent and independent manners. We illustrate the transition of the prostate tumor from AR signaling-dependent towards PI3K-AKT-mTOR pathway-dependent. Moreover, therapeutic strategies with inhibitors targeting PI3K-AKT-mTOR signal used in clinic and ongoing clinical trials are reviewed.

## 2. PI3K-AKT-mTOR Signaling

### 2.1. PI3K

PI3K, or phosphatidylinositol-3-kinase, is a plasma membrane-associated protein kinase that serves as a junction between the upstream growth factor/cytokine signals and the downstream intracellular signal transduction [11]. Among the three classes (I–III), class IA is most commonly linked to human cancer, including PCa [17]. This class of PI3K (referred to later as PI3K) is a heterodimer, which is formed by two functional subunits, a catalytic subunit (p110α, β, or δ isoform) and a regulatory subunit (p85α, p55α, p50α, p85β, or p55γ isoform) [18]. The catalytic subunit p110β is suggested to be the most relevant isoform to PCa progression and resistance due to an association with basal AKT (protein kinase B) activation in PCa models [17,19,20].

Usually, an activation of PI3K is mediated through receptor tyrosine kinases (RTKs), however, G-protein-coupled receptors and oncogenes, such as small GTPase RAS, can also activate PI3K, depending on the interaction context and the specificity towards PI3K subunits [21,22,23,24]. Once activated, PI3K phosphorylates phosphatidylinositol-4,5-biphosphate (PI(4,5)P_2_) to generate phosphatidylinositol-3,4,5-triphosphate (PIP_3_). In addition, PIP_3_ can be further converted to phosphatidylinositol-3,4-biphosphate (PI(3,4)P_2_) by Src homology 2 (SH2) domain containing inositol polyphosphate 5-phosphatase 1/2 (SHIP1/2) [25]. 

On the one hand, PIP_3_ activates intracellular signaling by recruiting and binding to a variety of proteins, including phosphoinositide-dependent kinase 1 (PDK1) and AKT, that contain pleckstrin homology (PH) domain [26]. This recruitment leads to the phosphorylation of AKT by PDK1 resulting in AKT activation. On the other hand, phosphorylation and activation of AKT by PDK1 can be facilitated by PI(3,4)P_2_ [25]. Notably, PIP_3_ can be inhibited via dephosphorylation by the tumor suppressor PTEN [27], whereas PI(3,4)P_2_ can be inhibited by both PTEN and INPP4B [28]. In PCa, loss or inactivation of PTEN and INPP4B seems to be common [17,18,29,30].

### 2.2. AKT

AKT belongs to a family of serine/threonine protein kinases and is the most famous downstream effector of the PI3K. Yet, AKT can also be activated by other kinases independent of PI3K signaling, such as by IKKε, SRC, ACK1, TANK binding kinase 1, DNA-dependent protein kinase, and ATM [31], suggesting multiple crosstalk situations in the tumor cells. Activation of AKT has been shown to drive PCa formation in vivo [11,32]. Moreover, phospho-proteomic analysis showed that AKT was commonly found to be active in metastatic tumor samples collected from rapid autopsy [17,33]. AKT is fully activated when it is phosphorylated at both Thr308 and Ser473 sites. However, phosphorylation at either site alone is also sufficient for AKT to partially mediate a subset of downstream cellular signaling [34,35]. Activated AKT regulates multiple cellular processes by phosphorylating several targets, including TSC2, GSK3, FOXO, ASK1, IKKα, CHK1, p27, p21, RAF1, BAD, MDM2, PRAS40, eNOS, AMPK, and WNK1 [36]. These downstream effectors link AKT activity to control protein synthesis, transcription, cell survival, apoptosis, proliferation, autophagy, and metabolism.

### 2.3. mTOR

Mammalian target of Rapamycin or mTOR is a serine/threonine protein kinase and is one of the major downstream effectors of AKT signaling. Interestingly, mTOR is expressed at higher levels in PCa compared to benign samples [17,37]. mTOR interacts with different proteins and forms two distinct complexes, mTORC1 and mTORC2. mTORC1 is comprised of mTOR, mLST8, DEPTOR, TTI1, TEL2, RAPTOR, and PRAS40, while mTORC2 is a complex of mTOR, mLST8, DEPTOR, TTI1, TEL2, RICTOR, mSIN1, and PROTOR1/2 [18]. Notably, mTORC1 is sensitive towards Rapamycin-mediated inhibition, whereas mTORC2 is not.

The activation of mTORC1 signaling is triggered by AKT-mediated phosphorylation of TSC2, which inhibits the TSC1/2 complex, leading to an activation of mTORC1 activator GTP-bound RHEB [38]. Moreover, mTORC1 repressor PRAS40 (also in the complex) is inhibited by AKT-mediated phosphorylation [39,40,41]. Notably, AMPK, GSK3, WNT, and energy signals can regulate TSC2 as well, linking the TSC2-mTORC1 pathway with other signaling cascades [42,43]. Protein synthesis is the major biological process mediated by activated mTORC1 signal via phosphorylation and activation of p70S6 kinase (p70S6K) [44] and inhibition of 4EBP1 [45]. Activated mTORC1 also blocks autophagy by inhibiting the autophagy inducing complex ULK1/ATG13/FIP200 [46,47,48]. Other biological processes controlled by active mTORC1 include lipid synthesis, energy metabolism, and lysosome biogenesis [38].

The activity of mTORC2 can be regulated by PI3K, RAS, AMPK, WNT, TSC1/2, and p70S6K [49,50,51,52,53]. Notably, inhibition of mTORC2 activity by p70S6K leads to a negative feedback regulation of the PI3K-AKT pathway, as mTORC2 facilitates AKT activation by phosphorylating Ser473 [54]. In contrast to biological processes controlled by mTORC1, active mTORC2 can phosphorylate several downstream effectors leading to cell survival, cell cycle progression, and actin remodeling. Moreover, it has been suggested that mTORC2 is required for the development of PCa lacking PTEN [55]. In line with this, the knockdown of PDK1 does not counteract enhanced PCa growth in PTEN-deficient transgenic mice [56], reflecting the possibility of mTORC2-mediated AKT and/or compensatory cascades activation.

Given the fact that the PI3K-AKT-mTOR pathway plays a critical role in controlling pro-survival cellular signals, it is possible that cancer cells, under therapeutic pressures, adaptively hyperactivate the pathway and its downstream cascades to compensate/overcome cellular stress. Pro-survival in general inhibits apoptosis, quiescence, and senescence, while promotes cell cycle progression. This may lead to even more aggressive cancer progression. In line with this, enhanced activity of PI3K-AKT-mTOR pathway correlates with PCa progression in the clinic [37,57,58,59,60]. Since feedback mechanisms activate pro-survival pathways, monotherapies using inhibitors of the PI3K-AKT-mTOR pathway could be limited in their efficacy.

## 3. PI3K-AKT-mTOR Interplays in Genomic and Non-Genomic AR Signaling

### 3.1. AR and Genomic Signaling

Inhibition of AR signaling is a major therapeutic aim in PCa therapy. Interestingly, the PI3K-AKT-mTOR pathway is well-documented for its crosstalk with AR signaling. Importantly, it may serve as a junction between genomic and non-genomic AR signaling.

AR is a transcription factor in a family of nuclear hormone receptor proteins. In general, inactive ARs reside in the cytoplasm of the cell and interact with chaperones and co-chaperones, such as heat shock proteins (HSPs) [61]. This interaction prevents AR from entering the nucleus. Canonically, initiation of the AR signaling begins with the binding of androgen to the ligand binding domain (LBD) of AR, which leads to a conformational change and AR activation (Figure 1). Hence, it can dissociate from chaperones, dimerize with another activated AR molecule, translocate into the nucleus, and bind to canonical androgen response element (ARE) for the regulating transcription of target genes. This is considered as “genomic AR signaling” and is ligand dependent, which is thought to occur over several hours [62]. Also, the activated AR can bind to non-canonical chromatin sites, with enhanced binding especially in cases of therapy resistance [63,64]. 

In the absence of androgens, genomic AR signaling can be regulated by cytoplasmic cascades, including the PI3K-AKT pathway, which are activated by various growth factors and cytokines [11,65,66,67] (Figure 1). In such cases, AR undergoes posttranslational modifications including phosphorylation, acetylation, methylation, ubiquitination, and SUMOylation [68]. The AR activated by transduction factors will eventually translocate into the nucleus and regulate target gene transcription.

### 3.2. AR and Non-Genomic Signaling

AR signaling can also occur in a non-genomic manner triggered by androgen-bound AR (Figure 1), which has been shown to occur rapidly within minutes [69,70]. In non-genomic AR signaling, nuclear translocation and DNA binding of AR are not required. The activated cytoplasmic AR rather interacts with or functions through other molecular effectors in the cytoplasm or at the membrane lipid rafts [62,71,72,73,74,75]. This activates multiple signaling cascades leading to cell proliferation, survival, anti-apoptotic, and migration. 

Several signaling molecules that are known to interact with AR include PI3K, AKT, SRC, RAS/RAF, PKC, MAPK/ERK, etc. [62,76]. Interestingly, these molecules, such as AKT, may serve as a junction between genomic and non-genomic AR signaling (Figure 1). Cross-phosphorylation events between AKT and AR are an example. On one hand, androgen-bound AR leads to an increase of AKT phosphorylation [77,78]. On the other hand, AKT phosphorylates AR at Ser210, Ser213, Ser215, Ser791, and Ser792 in order to regulate AR transcriptional activity and expression [68,79,80]. Thus, the downstream targets of those activated cascades triggered by non-genomic AR signaling may as well interact with non-ligand-bound AR and eventually lead to an overlapping scheme of ligand-independent genomic AR signaling. This suggests that AR signaling can be compensated/overcome easily when the cell encounters some selective pressures. 

## 4. PI3K-AKT-mTOR in PCa Progression and AR-Targeted Therapy Resistance

Application of therapeutic strategies against PCa depends on stages and particular situations of the disease. Also, the level of PCa diagnostic marker, prostate-specific antigen (PSA), is included in consideration. Surgery to remove the prostate gland (radical prostatectomy) may be the most effective option to remove the localized PCa that has low risk of progression to metastatic disease [81]. However, in some cases, due to the age, health issue, and refusion of patients, external beam radiation plus hormone therapy is rather chosen. 

Localized PCa as well as the early stage of advanced/metastatic PCa are generally androgen- and castration-sensitive (CSPCa) (Figure 2; left panel). Huggins and Hodges’s clinical observation in 1941 suggested that PCa growth can be controlled by reducing the level of androgens through castration [82]. Although basal activity of the PI3K-AKT-mTOR pathway should exist, the castration-sensitive characteristic indicates that these tumors are mainly relying on AR signaling. This observation brought up the current treatment option of androgen deprivation therapy (ADT) as a standard systemic hormone therapeutic strategy for PCa. ADT aims to lower serum testosterone concentration as much as possible in order to minimize stimulation of PCa cells [83]. ADT inhibits androgen production by blocking the hypothalamic-pituitary-testis feedback system with luteinizing hormone-releasing hormone analogues, also termed chemical castration. Adrenal ablating drugs are also used to decrease androgen synthesis from steroid precursors in the adrenal gland via inhibiting cytochrome P450 enzymes [83]. Unfortunately, despite the initial effectivity of ADT, the tumor in many patients gradually develops into a stage of castration-resistant PCa (CRPCa) [84], which is usually at a metastatic level and a major cause of morbidity as well as mortality.

Several mechanisms have been proposed for an occurrence of CRPCa, including increased intratumoral androgen synthesis, mutations of AR, amplification/overexpression of AR, and crosstalk between AR and other signaling pathways such as PI3K-AKT-mTOR [85,86] (Figure 2; middle panel). It seems that the tumor at this stage still relies on AR signaling, which is adaptively activated via multiple mechanisms [6,7,8,9,10]. For example, the activation of AR by AKT-mediated phosphorylation seems to generally occur in a low-testosterone state [79,87]. Therefore, the inhibition of AR signaling with AR antagonists is still applicable. Hence, in order to achieve the fully blockade of AR signaling, AR antagonists are often employed along with ADT to treat both CSPCa and CRPCa [88,89]. First generation of AR antagonists, such as Flutamide and Bicalutamide, are used to treat CSPCa, whereas second generation AR antagonists, such as Enzalutamide, Apalutamide, and Darolutamide, are used against CRPCa. Yet, after a certain period of time with treatment, PCa will eventually develop a resistance against AR antagonists. 

Interestingly, in contrast to an objective of inhibiting AR signaling, an application of using the supraphysiological androgen level (SAL) to treat CRPCa patients has been proposed as an optional therapeutic strategy [90]. Although, monotherapy of testosterone treatment showed disadvantages [91,92,93], combination therapy of SAL with ADT in a so-called bipolar androgen therapy (BAT) provided promising outcomes in clinical trials [94,95]. Importantly, it seems this therapy beneficially re-sensitizes CRPCa to AR antagonist treatment [95]. It is hypothesized that a rapid cycling between SAL and a depleted androgen level interfere with PCa cell adaptation on AR expression [90,96]. 

On one hand, mutated AR within the LBD and AR splice variants that lack LBD may be accumulated in CRPCa under selective pressure with AR antagonist treatment [97], and thus, PCa becomes less sensitive to AR antagonists. On the other hand, PCa cells with no AR expression may also be accumulated [98,99], and thus, the tumor progression is no longer dependent on AR signaling, such as the neuroendocrine prostate tumor [100]. Those cells may rely more on other nuclear hormone receptor signaling [101,102,103], or they may activate other AR independent signaling cascades [18,104,105] (Figure 2; right panel). In such cases, BAT also may not be effective anymore. Other therapeutic options that are not targeting AR may be applied during this stage of disease including chemotherapy, signaling pathway inhibitors, DNA damage repair pathway inhibitors, and immunotherapy [106,107].

Notably, after each therapeutic resistance stage from localized CSPCa to metastatic CRPCa, it seems that the tumor becomes more aggressive through activated adaptive responses, relies more on those alternative pathways, and becomes less dependent on AR signaling. This includes hyperactivation and deregulation of PI3K-AKT-mTOR signaling. It has been reported that about 42% of localized-stage and 100% of advanced-stage exhibit a deregulated PI3K-AKT-mTOR signaling pathway [11,18,108,109,110]. 

## 5. PI3K-AKT-mTOR Signaling Pathway as Resistance Mechanism to Therapy of PCa

### 5.1. Deregulation of PI3K-AKT-mTOR Signaling in PCa

An oncogenic role of PI3K-AKT-mTOR signaling as well as common genetic alterations in this pathway are well-documented [18]. Thus, it is not surprising that the deregulation of this pathway would mediate resistance against therapy and support the tumorigenesis of PCa.

#### 5.1.1. PTEN Loss of Function

One of the most common genetic alterations in prostate malignancies is the loss of tumor suppressor PTEN, which acts as a gatekeeper of the PI3K-AKT-mTOR pathway by dephosphorylating PIP_3_ back to PI(4,5)P_2_, leading to the inhibition of cell growth [17,18,29,111]. Inactivation of PTEN by deletion or mutations correlate strongly with enhanced PI3K-AKT-mTOR signaling, high Gleason score, and poor prognosis in advanced PCa [60,112,113,114]. It may occur in up to 50% of CRPCa cases [115]. Functional studies in vivo suggest *PTEN* loss as a genetic driver of murine prostate epithelium to become aggressive and locally invasive PCa, which has the ability to eventually acquire castration-resistance characteristics [116,117,118,119,120]. Importantly, PTEN loss links the development of CRPCa with androgen insensitivity, and perhaps also with AR antagonist insensitivity. It has been shown that the loss of PTEN de-represses EGR1 and c-Jun, which are negative regulators of AR activity, leading to suppression of the transcription of androgen-responsive genes [118,121]. Yet, the loss of PTEN still promotes cell growth. These findings reflect that the loss of PTEN, which leads to the activation of PI3K-AKT-mTOR, could still promote PCa growth without the need of androgen to transactivate AR target genes, such as in case of ADT. Furthermore, studies from both in vivo and in vitro of PCa have shown that the loss of AR or treatment with AR antagonist Enzalutamide leads to enhanced AKT signaling [78,118,121], supporting an activation of PI3K-AKT-mTOR signal to resist AR targeted therapy.

#### 5.1.2. PI3K Gain of Function

Other deregulation of the PI3K-AKT-mTOR pathway along with PTEN loss includes gain of function of PI3K, AKT, and/or mTOR themselves either by mutation or amplification. As an upstream effector of the pathway, mutations in *PIK3CA* encoding catalytic subunit p110α of PI3K is more common than the other isoforms. It has also been reported that approximately 30% of CRPCa patients harbor p110α mutations [117]. Consistently, Pearson et al. (2018) has summarized across nine PCa genomic datasets and showed that up to 28% of cases exhibit high-level amplification [114]. Although an anti-proliferative as well as anti-tumor activity has been observed with p110α-specific PI3K inhibitors [122,123,124], the question remains whether p110α is the major target in PCa therapy. This is due to a study of *Pten* loss-induced prostate tumor formation in mice, showing that the genetic deletion of *Pik3cb* encoding p110β isoform, but not deletion of *Pik3ca*, inhibits tumorigenesis together with reduction of Akt phosphorylation [19]. Thus, it suggests that p110β and not p110α may be more important isoform in PCa. Unlike p110α and β, the mutation and amplification of the p110δ isoform are infrequently detected in PCa patients [18]. Yet, the inactivation of p110δ inhibited PI3K-AKT signaling as well as cell proliferation in p110δ-highly expressed CRPCa cells [125], suggesting a potential target for a certain PCa subpopulation. In addition to alterations of the PI3K catalytic subunit, either genetic alterations or deletion of a regulatory subunit such as p85α may in part promote PI3K-AKT-mTOR signaling. On the one hand, the p85 regulatory subunit of PI3K suppresses p110 activity in the absence of stimuli, and on the other hand, p85-p110 heterodimerization is required after RTK activation for stimulating PI3K-AKT-mTOR signaling [18,22]. Importantly, it has been reported that both the p110β catalytic subunit and the p85α regulatory subunit are also essential for AR transactivation and PCa progression [126].

#### 5.1.3. AKT Gain of Function

Moving down to the key downstream effector of PI3K, gain of AKT function supports oncogenic, pro-survival/anti-apoptotic, and therapeutic resistant roles of this factor in PCa. Activation of AKT leads to the inhibition of cell death in the human CSPCa LNCaP cell line as well as promotes tumor growth and castration-resistance in transgenic mice [32,127]. Interestingly, gain of function of AKT seems to associate more with a high-level of *AKT* amplification than a mutation of *AKT*, since *AKT* amplification is more frequently detected [18]. In line with this, genetic alterations of well-known AKT regulators such as PDK1, PHLPP1/2, and PP2A may contribute also to the gain of AKT function [128,129]. An activation of AKT can be detected by the phosphorylation levels of Ser473 and Thr308. Surprisingly, either AR agonist at SAL used in BAT or the AR antagonist Enzalutamide used in clinic not only inhibit cell proliferation, but also enhance AKT phosphorylation in a cell culture model [78], rendering human CSPCa LNCaP cells to become apoptotic resistant. This in vitro model may represent an initial development of AR-independent CRPCa. 

One underlying mechanism of AR antagonist-induced AKT phosphorylation could be the regulation of *FKBP5* transcription, which is an AR-transactivated target. As an HSP90-associated co-chaperone that regulates the responsiveness of steroid hormone receptors, FKBP5 has been shown to negatively regulate AKT signaling by stabilizing the protein phosphatases PHLPP1/2 [121,130]. This means suppression of AR-mediated transactivation of *FKBP5* enhances AKT phosphorylation. It is also suggested that ADT stimulates AKT signaling for prostate tumor survival partly through *FKBP5* downregulation [121,131]. Hence, reduced expression of FKBP5 during ADT or AR-targeted therapy would lead to enhanced AKT activation and may facilitate the resistance to those therapies. In line with this, an increased level of AKT phosphorylation correlates with high Gleason grade and poorer survival in CRPCa [57,132,133]. Moreover, in combination with PTEN loss and high Gleason score, increased phosphorylation level of AKT can also be used to predict biochemical recurrence after radical prostatectomy [134].

##### 5.1.4. mTOR Gain of Function

Genetic alterations of mTORC1/2 upstream regulators TSC1/2 can lead to mTOR gain of function. On one hand, deletion of *Tsc1* in murine prostate epithelium causes prostate neoplasia, which is associated with elevated mTORC1 signaling [135]. On the other hand, the combined loss of heterozygous *Tsc2*^+/−^ and *Pten*^+/−^ is sufficient to promote mTOR activation and prostate tumorigenesis in vivo [136]. Apart from upstream regulators, genetic alterations of mTORC1/2 components themselves should not be overlooked. Interestingly, although the frequency of genetic alterations of other components is low in PCa, *DEPTOR* gene amplification appears to be frequently detected in up to 21.4% of cases and correlates with worse disease/progression-free survival [18]. This is surprising, since DEPTOR is actually an endogenous suppressor of mTOR kinase activity. One possible explanation would be that DEPTOR upregulation suppresses the feedback inhibition of mTORC1-p70S6K signal towards mTORC2 and PI3K, resulting in turn in an increased activation of AKT and mTORC1-independent functions [137,138,139]. Another important factor is RICTOR, a main component of mTORC2, which makes mTORC2 insensitive to Rapamycin. It has been reported that Rictor is required for prostate tumorigenesis induced by *Pten* loss in mice [55]. This is supported by in vivo studies showing that loss of Rictor suppresses *Pten*-deleted PCa growth. Therefore, it suggests that the oncogenic alteration of RICTOR that causes mTORC2 gain of function could potentially facilitate PCa progression. 

### 5.2. PI3K-AKT-mTOR as a Pro-Survival/Anti-Apoptotic Signaling

The most important key player for the pro-survival/anti-apoptotic role of PI3K-AKT-mTOR pathway appears to be AKT. It exerts multiple mechanisms to control cell survival and apoptosis by interacting directly with proteins of the apoptotic pathway or by regulating transcription factors that transcribe apoptotic-controlled genes [140].

One of the direct substrates of AKT is BAD, which is a pro-apoptotic factor and a member of the BCL-2 family of proteins that binds to and inhibits anti-apoptotic factor BCL-2 or BCL-XL. Inactivation of BAD occurs through phosphorylation at Ser136 by active AKT [141], leading to an activation of BCL-2 or BCL-XL and cell survival. In CRPCa, it has been reported that PI3K-AKT signaling is activated by the clinically used AR antagonist Enzalutamide, facilitating cells to evade apoptosis via BAD inactivation [142]. This suggests one possible mechanism of how CRPCa cells become AR antagonist resistant.

Other apoptotic pathway regulators that interact and become phosphorylated by AKT are the cell death protease Caspase-9 and apoptosis signal-regulating kinases ASK1, MLK3, and SEK1 [140]. AKT phosphorylates Caspase-9 at Ser19 and inhibits its protease activity [143]. AKT phosphorylates on Ser83 of ASK1, Ser674 of MLK3, and Ser78 of SEK1, causing these factors to be inactive [144,145,146,147]. These AKT activities result in the inhibition of apoptosis induction and promotion of cell survival.

In PCa, AKT-mediated ASK1 inhibition may be critical for cell survival. It is suggested that the HSP90 chaperone is required for AKT-mediated phosphorylation and inhibition of ASK1 [145]. Interestingly, the HSP90 inhibitor sensitizes CSPCa LNCaP cells to apoptosis under the treatment with SAL used in BAT [78]. In line with this, SAL-treated LNCaP cells, although at the growth arrest stage, exhibit apoptotic resistance by enhancing the AKT-mTOR pathway. In CRPCa and AR antagonist-resistant CRPCa, AKT-mediated ASK1 also seems to be important. It has been shown in C4-2 as well as PC3 cells that disabled homolog 2-interacting protein (DAB2IP) coordinates both PI3K-AKT and ASK1 pathways for cell survival and apoptosis [148]. Gain of function of this protein can suppress the PI3K-AKT pathway and enhance ASK1 activation, leading to cell apoptosis, whereas loss of function of DAB2IP leads to opposite effects. Loss of DAB2IP has also been shown to accelerate PCa growth in vivo [148]. Notably, loss of DAB2IP is often detected in androgen-independent PCa [149,150].

Apart from directly regulating the proteins of the apoptotic pathway, AKT is able to regulate cell survival and anti-apoptosis via transcription factors. This includes phosphorylation of FOXO1/2/3/4, IκB kinase, MDM2, CREB, YAP, and AR. By phosphorylating FOXO, IκB kinase, and CREB, AKT indirectly controls transcription machinery of apoptosis regulating genes by suppressing pro-apoptotic factors of the BCL-2 family, as well as promoting anti-apoptotic factors of the BCL-2 family and caspase inhibitors [140]. Phosphorylated YAP by AKT suppresses apoptosis mediated by p73-regulated pro-apoptotic gene transcription [151]. Moreover, AKT-induced phosphorylation and translocation of MDM2 from the cytoplasm to the nucleus. Thus, AKT results in an inactivation/degradation of p53 and thereby antagonizes p53-mediated pro-apoptotic transcriptional regulation [152,153,154].

Interestingly, the PI3K-AKT pathway inhibits not only apoptosis but also triggers G_1_/S cell cycle progression. It is known that AKT phosphorylates and inhibits GSK3β to prevent cyclin D1 degradation [155]. Moreover, the PI3K-AKT pathway also directly affects the cell cycle inhibitor p21^Waf1/Cip1^ and p27^Kip1^ by phosphorylation that causes cytoplasmic accumulation and inhibition of the access to the cyclin-CDK targets [156,157].

As a downstream effector of the PI3K-AKT-mTOR pathway, mTORC1/2 are also involved in cell survival and anti-apoptosis regulation [14]. Via multiple signaling cascades, including AKT activation as described in previous sections, mTORC2 is well-linked to a role regarding cell survival/anti-apoptosis [158,159,160]. Unlike mTORC2, mTORC1 is more well-known for its role in ribosomal biogenesis and protein translation, yet these are important processes for cancer cell survival and proliferation [161]. 

Several studies have implicated mTORC1 in apoptosis and cell survival regulation [162,163,164,165]. For example, it has been described that mTOR regulates cell survival after etoposide treatment in acute myeloid leukemia cells (AML) [162]. Furthermore, a recent study showed that inhibition of mTORC1 improved the killing of AML cells by chemotherapy in a time-specific manner [165]. In mouse embryonic fibroblasts (MEFs), mTORC1 can control mitochondrial dynamics and cell survival via MTFP1 [164]. In PCa cells, SAL-treated cells that are apoptosis resistant are accompanied by enhanced phosphorylation of both AKT and ribosomal S6 proteins [78], suggesting an enhanced mTORC1 activity under SAL condition. In line with this, inhibition of AKT does not abolish SAL-induced S6 phosphorylation, suggesting an AKT bypassing mechanism by activated AR. Interestingly, the phosphorylation state of ribosomal S6 protein, a target of mTORC1 downstream effector p70S6K, is implicated with cell survival [166,167,168]. MEFs carrying S6^P-/-^ are sensitive to TRAIL-, etoposide-, and MG132-induced apoptosis [167,168]. This suggests that several factors within the PI3K-AKT-mTOR pathway possess, individually in an autonomous manner, pro-survival activity.

## 6. Targeting PI3K-AKT-mTOR Signaling in PCa 

Since the PI3K-AKT-mTOR pathway is commonly activated in advanced stages of PCa, this pathway may represent a potential target to effectively inhibit PCa growth and to overcome resistance of AR targeted therapy [11,169]. Many inhibitors for this pathway have been tested as monotherapy or in combination with other agents in preclinical and clinical trials as discussed below and summarized in Table 1.

### 6.1. PI3K Inhibitors

Inhibitors of PI3K can be divided/classified into pan-PI3K inhibitors and isoform-specific PI3K inhibitors. Pan-PI3K inhibitors target the catalytic subunit of all three isoforms of class IA PI3K. One potent oral pan-PI3K inhibitor is BKM120, which suppressed tumor growth in PC3-xenograft mouse model [170]. An IC_50_ of BKM120 at 3.23 µM and 2.81 µM has been documented for human CSPCa LNCaP cell line and AR antagonist-insensitive CRPCa PC3 cell line, respectively [171]. The inhibitor showed evidence of partial response in one out of 21 patients in a phase I first-in-men study, and seven patients remained on treatment for ≥8 month [172]. BKM120 is currently under investigation in metastatic CRPCa in a phase II study. PX866 is a synthetic derivative of wortmannin, which covalently binds to Lys802 in the ATP catalytic site of the PI3K [173]. In a phase II study PX866 was well tolerated in patients with recurrent or metastatic CRPCa, and 14 of 25 patients were progression-free at 12 weeks [174]. However, the clinical use of pan-PI3K inhibitors in monotherapies is limited by the compensatory increase in AR signaling [121].

Isoform-specific PI3K inhibitors, such as BYL719 and MLN1117, aim to specifically target the p110 isoform to decrease side effects like insulin resistance and hyperglycemia. Targeting p110 might be an effective treatment option since *PIK3CA*, the gene that encodes p110α, is commonly altered in metastatic PCa. In line with this, both inhibitors indicated antiproliferative and antitumor activity in cell lines and xenograft models with *PIK3CA* mutations [122,123]. However, compensatory effects by other PI3K isoforms can occur and will eventually lead to an activation of other pro-survival pathways including re-activation of AR-signaling. This highlights the need to target, in addition to blockage of androgen action, the androgen-independent receptor activation as well as pro-survival cancer cell pathways.

### 6.2. AKT Inhibitors

As key regulator of pro-survival pathways, AKT provides an attractive target for therapeutic interventions. AKT inhibitors can be categorized based on their mechanism of action into six classes [175]. The first class comprises ATP-competitive inhibitors of AKT such as Ipatasertib and AZD5363 [176]. This kind of inhibitors, unlike allosteric inhibitors, lead to hyperphosphorylation of AKT at the Thr308 and Ser473 residues. Lipid-based AKT inhibitors that prevent the generation of PIP_3_ by PI3K represent the second class. Different phosphatidylinositol analogs, such as Calbiochem AKT inhibitors (Sigma Aldrich, Cat.-Nr.: 124005, St. Louis, MO, USA ), and PI3K inhibitors, such as PX-866, use this mechanism of action [177]. The third class consists of compounds named pseudo-substrate inhibitors, including AKTide-2 T and FOXO3 hybrid [178,179]. The fourth class is made of allosteric inhibitors of the kinase domain of AKT, such as MK-2206. Importantly, antibodies can also be used to inhibit AKT activity and thus represent the fifth class. The last class of inhibitors targets the PH-domain of AKT to interfere with the translocation of AKT to the plasma membrane and therefore blocking AKT phosphorylation and activation. This class includes compounds such as PX-316 [175]. Despite the large number of AKT inhibitors only allosteric and ATP-competitive AKT inhibitors have so far reached the clinical phase for PCa.

Importantly, preclinical studies with the allosteric AKT inhibitor Perifosine reduced proliferation and induced apoptosis and differentiation in PC3 [180] and PTEN-deficient PCa cells [181]. In LNCaP and PC3 cells, an IC_50_ of Perifosine is ~5 µM, whereas DU145 cells exhibit an IC_50_ value of 15 µM [181]. Perifosine is an alkylphospholid that accumulates in cell membranes; however, besides the AKT inhibitory properties, the exact mechanism of action of this inhibitor remains unknown [17]. In men with CRPCa, Perifosine was well-tolerated but lacked evidence of a radiographic or PSA response [182]. Another clinically tested allosteric inhibitor is MK-2206 [176]. This inhibitor predominantly targets AKT1 (IC_50_ = 5 nM) and AKT2 (IC_50_ = 12 nM) with lower potency against AKT3 (IC_50_ = 65 nM) [183]. MK-2206 has in vitro and in vivo antitumor activity [183].

Recent studies suggest that active site AKT inhibitors may have greater antitumor activity. In line with this hypothesis, the active site inhibitor AZD5363 was reported to suppress proliferation and to increase apoptosis in PCa cell lines and the LNCaP xenograft model [184]. An IC_50_ of AZD5363 in LNCaP cells is at nanomolar concentrations [185]. Moreover, a report from a phase I study has suggested that the combination of AZD5363 and AR antagonist Enzalutmide is tolerable and has antitumor activity [186]. The sensitivity towards AZD5363 is correlated with the presence of *PIK3CA* mutations, AKT1^E17K^ mutations, or PTEN loss [185]. Especially, the combination of AZD5363 with Docetaxel led in 70% of men with metastatic CRPCa to > 50% reduction of PSA levels illustrating the potential of active site inhibitors in combination therapy [187].

The optimization of ATP-competitive AKT inhibitors led to the development of Ipatasertib (GDC-0068), a highly selective AKT1-3 inhibitor with an IC50 value of 5, 18, and 8 nM, respectively. Currently, Ipatasertib is evaluated in a phase 3 trial in combination with Abiraterone and prednisolone in metastatic CRPCa (NCT03072238). Importantly, the pan-AKT inhibitor GSK2141795 (Uprosertib) showed measurable responses in a phase I study of seven patients, while six men had a stable disease [188]. Moreover, Uprosertib as monotherapy was reported to be safe and well-tolerated at the recommended phase II dose study [189]. These results demonstrate the clinical potential of AKT inhibitors in a special subset of patients. However, a major issue of many AKT inhibitors is the relief of negative feedback inhibition and activation of different RTKs [190]. Hence, inhibitors have been developed that target the downstream mediators of AKT.

### 6.3. mTOR Inhibitors

The mTOR is an important downstream effector of the PI3K pathway, which integrates extracellular signal transduction with metabolic processes representing a potent target to control cellular growth. The allosteric mTORC1 inhibitors Rapamycin and its analogs, including Everolimus and Temsirolimus (Table 1), were the first PI3K/AKT/mTOR pathway inhibitors that were assessed in clinical trials [169]. Pre-clinical studies of mTORC1 inhibitors were promising and reverted prostatic intraepithelial neoplasia (PIN) in mouse models overexpressing AKT [191]. However, clinical trials using single mTORC1 inhibitors lacked favorable clinical responses [192,193]. Rapamycin demonstrated successful inhibition of the mTORC1 target phospho-S6 in patients with intermediate to high-risk PCa, but no significant effects on tumor cell proliferation, induction of apoptosis, or PSA levels were observed [194]. This may be due to mTORC2 mediated compensation and activation of AKT [169,195]. Further, Rapalogs showed incomplete inhibition of downstream effectors, including EIF4E, since Rapalogs do not bind directly to and inhibit the catalytic core of the mTOR kinase. Instead, they bind to FKBP12 to allosterically inhibit mTOR [196].

Next, a new class of mTOR inhibitors blocking mTORC1 and mTORC2 activity have been developed with the aim to prevent the feedback induction of AKT. The dual mTORC1/2 inhibitors, such as MLN0128 and AZD2014, demonstrated improved efficiency due to a more potent inhibition of downstream targets like 4EBP1, protein synthesis and induction of cell cycle arrest in different cell lines [197]. In addition, MLN0128 prevented not only PCa invasion and metastasis, but further induced apoptosis [198]. This inhibitor has been previously tested in a phase II study in advanced CRPCa [199]. Interestingly, AZD2014 inhibited migration, invasion, and EMT progression in Docetaxel-sensitive and Docetaxel-resistant CRPCa cells more potent compared to Rapamycin. Note that, a much lesser dose of AZD2014 (0.204 µM) than Rapamycin (6.49 µM) is also sufficient to exhibit an IC_50_ in a human CRPCa C4-2 cell line [200]. Moreover, AZD2014 was more potent in the inhibition of 4EBP1 and AKT phosphorylation than Rapamycin [200]. This inhibitor was tested in men with high-risk PCa given prior to radical prostatectomy (NCT02064608). Although completed, there is no published report for the outcome of this clinical trial yet.

Notably, one limitation of dual mTORC1/2 inhibitors is the loss of S6K-mediated negative feedback regulation leading to activation of PI3K signaling via RTK activation [169]. This led to trials with dual PI3K and mTORC1/2 inhibitors.

### 6.4. Dual PI3K and mTORC1/2 Inhibitors

Dual PI3K and mTORC1/2 inhibitors target all four p110 isoforms and both mTOR complexes in order to achieve a more complete blockade of the PI3K-AKT-mTOR signaling axis [11]. In preclinical studies, GDC-0980 and BEZ235 inhibited proliferation of multiple cell lines and induced G1 arrest [201,202]. Moreover, GDC-0980 caused apoptosis in cell lines harboring *PIK3CA* mutations or *PTEN* loss, causing direct activation of the PI3K pathway [202]. IC_50_ of GDC-0980 with 0.036 µM and BEZ235 with 0.038 µM have been documented in LNCaP cells, whereas 0.2 µM GDC-0980 and 0.06 µM BEZ235 are sufficient to exhibit IC_50_ in PC3 cells [202,203]. Both inhibitors are well tolerated in the clinic with mild side effects, including nausea, vomiting, diarrhea, and fatigue [204,205,206]. Currently, GDC-0980 in combination with Abiraterone acetate is tested in CRPCa in phase I//II clinical trials (Table 1).

The limited efficacy of monotherapy with PI3K/AKT/mTOR inhibitors is caused by complex reciprocal feedback mechanisms that include the AR and interaction with other signaling pathways. These findings urged towards the need to develop combination therapies.

### 6.5. Combination Strategies with PI3K-AKT-mTOR Inhibitors

The activation of the PI3K-AKT-mTOR pathway is implicated in resistance to chemotherapy, for example, to docetaxel [207]. Mechanistically AKT is activated following chemotherapy-induced double strand breaks. Thus, AKT promotes the survival of cancer cells triggering anti-apoptotic pathways [208,209,210]. In line with this, inhibition of AKT has been indicated to hypersensitize cells to different chemotherapeutic agents in pre-clinical trials [185,211]. The results are supported by previous xenograft studies, showing that AZD5363 significantly enhances the activity of docetaxel [185]. Moreover, MK-2206 indicated synergistic antitumor efficacy with docetaxel in PC3 xenograft models [183].

Since it has been reported that a strong suppression of AR signaling by AR antagonists causes reciprocal activation of PI3K-AKT, dual inhibition with ADT and PI3K, AKT, or mTOR inhibitors may lead to more potent PCa growth inhibition [118,121]. In line with this, the combination of Everolimus and Bicalutamide significantly reduced tumor growth rates and tumor volume in LNCaP xenografts compared to Bicalutamide alone [212]. Moreover, this study suggests that combined targeting of AR and mTOR inhibitors can restore sensitivity to anti-androgen therapy. However, clinical results of Everolimus plus Bicalutamide in CRPCa have been contradictory. A phase II study reported low activity of the combination treatment [213]. In contrast, another study showed a response in 18 out of 24 patients (75%) treated with Everolimus and Bicalutamide [214].

Promising results from a phase I/II study using Ipatasertib in combination with Abiraterone demonstrated improved radiographic progression-free survival (PFS) and overall survival (OS) in patients with CRPCa previously treated with Docetaxel compared to single treatment [215]. Along with these observations, the PI3K inhibitor BEZ235 together with the AR antagonist Enzalutamide resulted in an enhanced apoptosis rate in a PTEN negative PCa [121].

Overall, the clinical application of monotherapy of the PI3K-AKT-mTOR pathway can be limited by drug-resistance, dose-limiting toxicity, and complex reciprocal feedback such as interaction with AR and other signaling pathways [216]. In addition, compensatory mechanism by other factors leading to maintenance of the pathway can also be the reason [217]. Thus, combined application of PI3K-AKT-mTOR inhibitors may have great potential for clinical benefit. Yet, due to complexity of cellular signaling network and unique individual characteristic, stratification of patients and particular biomarkers are needed to improve clinical efficiency of these inhibitors. Current clinically relevant biomarkers include the status of *PI3KCA*, *PI3KCB,* or *AKT* somatic alteration and *PTEN* loss of PCa patients [218]. However, further identification/characterization of new biomarkers for patient selection in the clinic is still required to enable the development of personalized therapy.

## 7. Conclusions

Several factors within the PI3K-AKT-mTOR signaling have pro-survival activity in PCa cells. These factors can even act individually and independently from each other. Thus, inhibition of one factor within this signaling cascade can lead to a feedback mechanism that results in alternative pro-survival bypass pathways. Therefore, a combinatorial treatment with the goal to inhibit the AR and more than one factor of the PI3K-AKT-mTOR signaling could be beneficial. However, optimally, a patient stratification must precede such a combinatorial therapy with suitable markers to detect which pro-survival factor is being activated in order to use appropriate inhibitors in an individualized manner. This would allow patient-oriented use of a target specific treatment for a personalized therapy.

## Figures and Tables

**Figure 1 ijms-22-11088-f001:**
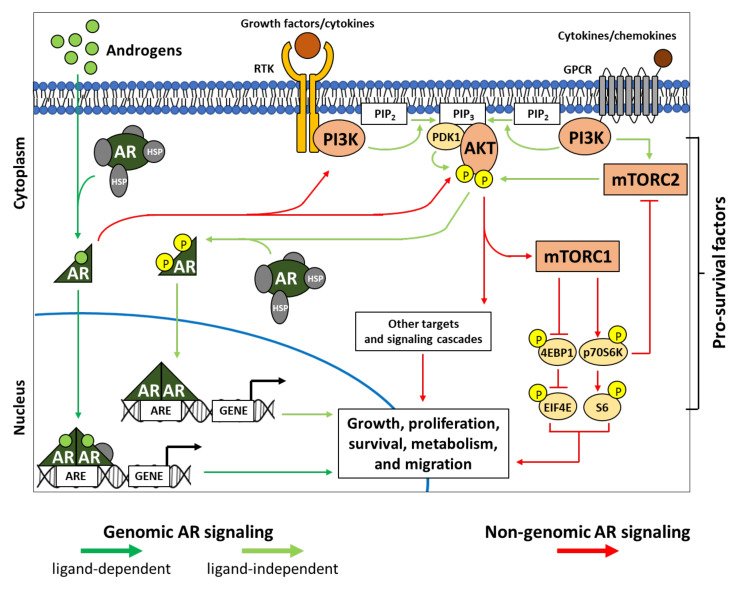
PI3K-AKT-mTOR interplays in genomic and non-genomic AR signaling. AR signaling can be classified into genomic and non-genomic signaling. Genomic AR signaling involves the translocation of activated AR into the nucleus, the binding to ARE of target genes, and the regulation of transcription activity. In contrast, non-genomic AR signaling does not require AR translocation and DNA binding. In general, AR activation is initiated from the binding of androgen to the receptor, which leads to conformational changes and dissociation of AR from chaperones such as HSPs. This is known as androgen- or ligand-dependent genomic signaling (dark green arrows). However, in the absence of androgens, AR can be activated via phosphorylation mediated by multiple cytoplasmic factors including AKT, a key biological processing factor of the PI3K-AKT-mTOR pathway. Thus, it is considered as ligand-independent genomic AR signaling (light green arrows). In turn, as non-genomic AR signaling (red arrows), the activation of AR can activate the PI3K-AKT-mTOR pathway by interacting with PI3K and AKT. Activation of PI3K-AKT triggers downstream effectors, including mTOR and other signaling cascades, leading to the promotion of growth, proliferation, survival, metabolism, migration, etc. AKT, protein kinase B; AR, androgen receptor; ARE, androgen response element; EIF4E, eukaryotic translation initiation factor 4E; GPCR, G-protein-coupled receptor; HSPs, heat shock proteins; mTORC1/2, mammalian target of Rapamycin complex 1/2; P, phosphorylation; PDK1, phosphoinositide-dependent kinase 1; PI3K, phosphatidylinositol-3-kinase; PIP_2_, phosphatidylinositol-4,5-biphosphate; PIP_3_, phosphatidyl-inositol-3,4,5-biphosphate; p70S6K, p70S6 kinase; RTK, receptor tyrosine kinase; S6, ribosomal S6 protein; 4EBP1, 4E binding protein 1.

**Figure 2 ijms-22-11088-f002:**
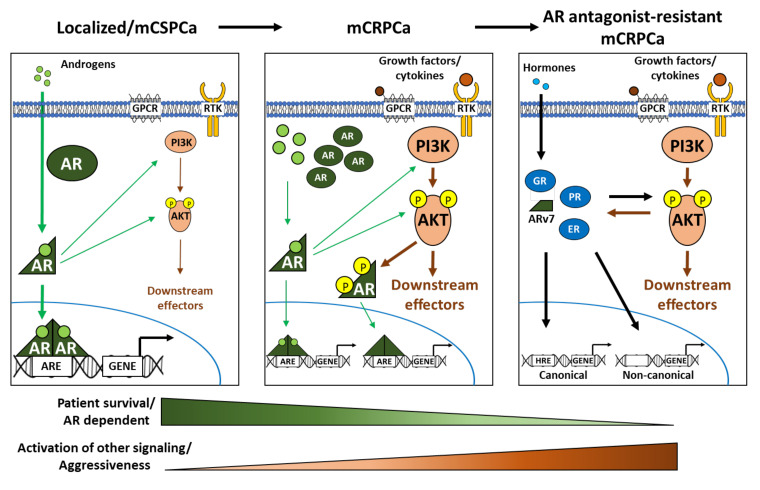
Transition from AR signaling-dependent to PI3K-AKT-mTOR pathway-dependent in PCa. PCa as a localized disease and at the early stage of metastatic disease is androgen- and castration-sensitive. The tumor at these stages strongly relies on systemic/circulating androgens for activating AR signaling to stimulate growth and progression (left panel). This makes it easy to suppress the tumor with therapeutic approaches that target androgen synthesis and AR signaling. However, after a certain period of time, the tumor will eventually develop a resistant stage called castration-resistance (middle panel). At this stage, ADT and the first generation of AR antagonists are not any more effective due to several hypothesized reasons, including intratumoral androgen synthesis, amplification of AR, activation of ligand-independent genomic AR signaling (e.g., via PI3K-AKT-mTOR), etc. These mechanisms suggest that the tumor at this stage still relies on AR signaling, although via adaptive responses, but could still be treated with second generation AR antagonists. Notably, bipolar androgen therapy (BAT) with cycling treatment of ADT and supraphysiological androgen levels seems to be also effective. Again, after a period of time, the tumor will become resistant to AR antagonist (right panel). Presumably, the therapeutic pressure by AR antagonists will selectively lead to the accumulation of PCa cells that no longer express AR or at least express mutated AR/AR splice variants (e.g., ARv7) lacking ligand binding domain. This makes the tumor insensitive to AR antagonists and BAT. Independent of AR, the tumor at this stage fully relies on hyperactivation of multiple cellular signaling cascades such as PI3K-AKT-mTOR and signaling of other nuclear hormone receptors (e.g., GR, PR, and ER). Along with more advance stages of the disease, an aggressiveness of the PCa is enhancing, whereas the survival of the PCa patient is reducing. AKT, protein kinase B; AR, androgen receptor; ARE, androgen response element; ARv7, androgen receptor splice variant 7; ER, estrogen receptor; GPCR, G-protein-coupled receptor; GR, glucocorticoid receptor; HRE, hormone response element; mCSPCa, metastatic castration-sensitive prostate cancer; mCRPCa, metastatic castration-resistant prostate cancer; P, phosphorylation; PI3K, phosphatidylinositol-3-kinase; PR, progesterone receptor; RTK, receptor tyrosine kinase.

**Table 1 ijms-22-11088-t001:** PI3K-AKT-mTOR pathway inhibitors used in clinical trials.

Target	Agent	Phase	Regimen	Population	Status	Registry
Pan-PI3K inhibitors	BKM120 (Buparlisib)	I	+Abiraterone acetate (CYP17A1 inhibitor)	CRPCa progressed on Abiraterone acetate	Completed	NCT01634061
		I	+Abiraterone acetate	Docetaxel -pretreated metastatic CRPCa	Terminated	NCT01741753
		II	Monotherapy	Metastatic CRPCa progressed following ADT and chemotherapy	Terminated	NCT01385293
		II	Monotherapy	High-risk, localized prostate cancer prior to radical prostatectomy	Terminated	NCT01695473
	PX866 (Sonolisib)	II	Monotherapy	Metastatic CRPCa progressed following ADT	Completed	NCT01331083
Dual PI3K/ mTOR inhibitors	BEZ235	I	+Abiraterone acetate	CRPCa progressed on Abiraterone acetate	Completed	NCT01634061
	GDC-0980	II	+Abiraterone acetate	Docetaxel pre-treated CRPCa	Active, not recruiting	NCT01485861
	LY3023414	II	+Enzalutamide	Metastatic CRPCa	Completed	NCT02407054
AKT inhibitors	AZD5363 (capivasertib)	I	Monotherapy	Metastatic CRPCa	Completed	NCT01692262
		I	+Enzalutamide or Abiraterone	Metastatic CRPCa	Completed	NCT04087174
		I/II	+Docetaxel and Prednisolone (glucocorticoid)	Metastatic CRPCa	Active, not recruiting	NCT02121639
	GSK2141795 (Uprosertib)	I	Monotherapy	Castration-resistant, locally advanced or metastatic with/without PTEN loss	Completed	NCT00920257
	MK2206	II	+Bicalutamide (anti-androgen)	PCa patients with biochemical relapse and rising PSA after primary therapy	Active, not recruiting	NCT01251861
		I	+Hydroxychloroquine	Stage III PCa	Active, not recruiting	NCT01480154
	GDC-0068 (Ipatasertib)	II	+Abiraterone acetate and Prednisone	Metastatic or advanced prostate carcinoma	Active, not recruiting	NCT01485861
		Ib	+Atezolizumab and Docetaxel	Metastatic CRPCa	Recruiting	NCT04404140
		III	+Abiraterone acetate + Prednisone/Prednisolone	Metastatic CRPCa	Active, not recruiting	NCT03072238
	Perifosine	II	Monotherapy	Metastatic androgen-independent PCa	Completed	NCT00060437
mTORC1 inhibitors	Everolimus	II	Monotherapy	Metastatic CRPCa	Completed	NCT00629525
		I	+Radiation therapy	Biochemical recurrence after radical prostatectomy	Completed	NCT01548807
		II	+Pasireotide (somatostatin)	Chemotherapy-naive CRPCa	Terminated	NCT01313559
		I/II	+Docetaxel, Bevacizumab (VEGF inhibitor)	Metastatic CRPCa	Completed	NCT00574769
		I/II	+Docetaxel	Metastatic CRPCa	Completed	NCT00459186
		II	+Carboplatin and Predisone	Metastatic CRPCa progressed after Docetaxel	Completed	NCT01051570
		II	+Bicalutamide	Recurrent or metastatic CRPCa after first-line ADT	Completed	NCT00814788
	Temsirolimus	I/II	+Bevacizumab	Chemotherapy-treated metastatic CRPCa	Completed	NCT01083368
		II	Monotherapy	Chemotherapy-treated metastatic CRPCa	Terminated	NCT00887640
		II	Monotherapy	Chemotherapy-naive metastatic CRPCa	Completed	NCT00919035
		I	+Vorinostat (HDAC inhibitor)	Metastatic CRPCa	Terminated	NCT01174199
		I/II	+Docetaxel	CRPC receiving first-line docetaxel	Completed	NCT01206036
		I/II	+Cixutumumab	Metastatic CRPCa	Completed	NCT01026623
Dual mTORC1/2 inhibitors	MLN0128	II	Monotherapy	Metastatic CRPCa	Completed	NCT02091531
	AZD2014	I	Monotherapy	High-risk PCa before radical prostatectomy	Completed	NCT02064608
		I	Monotherapy/+ Abiraterone acetate	CRPCa	Completed	NCT01884285

+ indicates co-treatment of agent and indicated regimen. ADT, androgen deprivation therapy; CRPCa, castration-resistance prostate cancer; PCa, prostate cancer.

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
