# Peer review of "Role of PI3K-AKT-mTOR Pathway as a Pro-Survival Signaling and Resistance-Mediating Mechanism to Therapy of Prostate Cancer"

_ijms, 2021, doi:10.3390/ijms222011088_

Round 1

Reviewer 1 Report

This manuscript highlights the role of PI3K-AKT-mTOR signaling as a mechanism of adaptive resistance to prostate cancer therapy. Furthermore,  therapeutic strategies to target this pathway are discussed. In general, the manuscript is well written, using concise and adequate English language. The background is clearly introduced to the reader and highlights how PCa becomes PI3K-dependent. Besides, there is sufficient evidence supporting the functional role of PI3K pathway in PCa specifically,  and the consequences of pathway deregulation. 

However, the authors could touch on the proposed mechanisms of resistance to PI3K inhibitors in clinical practice.

Furthermore, the authors should consider to refer to other relevant papers covering the role of PI3K in PCa,  such as the following: Braglia et al.  2020 BBA-MRC; Bertacchini et al. 2019  J Cell Physiol

Author Response

Reviewer #1

This manuscript highlights the role of PI3K-AKT-mTOR signaling as a mechanism of adaptive resistance to prostate cancer therapy. Furthermore,  therapeutic strategies to target this pathway are discussed. In general, the manuscript is well written, using concise and adequate English language. The background is clearly introduced to the reader and highlights how PCa becomes PI3K-dependent. Besides, there is sufficient evidence supporting the functional role of PI3K pathway in PCa specifically,  and the consequences of pathway deregulation. 

  1. However, the authors could touch on the proposed mechanisms of resistance to PI3K inhibitors in clinical practice.

Authors‘ response: We thank you the reviewer for the compliments and suggestion. We have now mentioned proposed mechanisms of resistance for PI3K-AKT-mTOR inhibitors. Please see section 6.5, lines 721-726.

  1. Furthermore, the authors should consider to refer to other relevant papers covering the role of PI3K in PCa,  such as the following: Braglia et al.  2020 BBA-MRC; Bertacchini et al. 2019  J Cell Physiol

Authors‘ response: We have now included the suggested references. Please see ref #216 and #217.

Reviewer 2 Report

 Some of my comments from my side are as follows - 

  1. The diagram must be revised carefully to make proper understanding for readers.
  2.  The author should give some docking studies to support the inhibition or binding capacity.
  3.  Author should provide some  in-vitro study data to support the study though its review. 

Author Response

Reviewer #2

Some of my comments from my side are as follows - 

  1. The diagram must be revised carefully to make proper understanding for readers.

Authors‘ response: We have now modified the figure. Please see figure 1.

  1. The author should give some docking studies to support the inhibition or binding capacity.

Authors‘ response: We thank you the reviewer for the suggestions. We have now included the IC50 of multiple inhibitors of PI3K-AKT-mTOR pathway. Please see point 6.1-6.4.

  1. Author should provide some  in-vitro study data to support the study though its review. 

Authors‘ response: Thank you the comment on this point. Throughout this review, we have provided supported information collected mostly from clinical, pre-clinical, and in vivo studies in which we considered that it should be more meaningful and relevant to clinics.

Reviewer 3 Report

The review by Pungsrinont et al described the role of PI3K-AKT-mTOR signaling as a resistance mechanism for PCa therapy and the related therapeutic strategies.

The manuscript is very interesting, linear and well-written.

To further improve the manuscript, I suggest to add:

1) a table summarizing clinical trials

2) a brief paragraph summarizing the state of the art of the biomarkers for personalized therapy.

Author Response

Reviewer #3

The review by Pungsrinont et al described the role of PI3K-AKT-mTOR signaling as a resistance mechanism for PCa therapy and the related therapeutic strategies.

The manuscript is very interesting, linear and well-written.

To further improve the manuscript, I suggest to add:

  1. a table summarizing clinical trials

Authors‘ response: We appreciate the reviewer for grateful compliments. We have summarized clinical trials of multiple inhibitors of PI3K-AKT-mTOR pathway in Table 1.

  1. a brief paragraph summarizing the state of the art of the biomarkers for personalized therapy.

Authors‘ response: Thank you for the reviewer suggestion. Several biomarkers have been mentioned in the main text. However, we have now additionally mentioned some biomarkers in the last paragraph of section 6.5, line 731-735.

Reviewer 4 Report

In the current review article, authors have done an elegant job reviewing the current knowledge of the role of PI3K-Akt-mTOR signaling pathway. The review article is very informative, yet not incremental from the existing papers. The article comprehensively reviewed the functions of each element of the signaling pathway individually, talked about the clinical implications and treatment stratification. The field will benefit tremendously from this paper.

Author Response

Reviewer #4

In the current review article, authors have done an elegant job reviewing the current knowledge of the role of PI3K-Akt-mTOR signaling pathway. The review article is very informative, yet not incremental from the existing papers. The article comprehensively reviewed the functions of each element of the signaling pathway individually, talked about the clinical implications and treatment stratification. The field will benefit tremendously from this paper.

Authors‘ response: We thank you the reviewer for grading this manuscript as an elegant job.